# The Effects of Number of Fire Dispatches and Other Situational Factors on Voluntary Exercise Training Among Korean Firefighters: A Multilevel Logistic Regression Analysis

**DOI:** 10.3390/ijerph17165913

**Published:** 2020-08-14

**Authors:** Junhye Kwon, Seiyeong Park, Chung Gun Lee, Wook Song, Dong-il Seo, Jung-jun Park, Han-joon Lee, Hyun Joo Kang, Yeon Soon Ahn

**Affiliations:** 1Department of Physical Education, College of Education, Seoul National University, Seoul 08826, Korea; 2017_25059@snu.ac.kr (J.K.); pseiy09@snu.ac.kr (S.P.); songw3@snu.ac.kr (W.S.); 2Institute of Sport Science, Seoul National University, Seoul 08826, Korea; 3Institute on Aging, Seoul National University, Seoul 08826, Korea; 4Department of Sport Science, College of Liberal Arts, Dongguk University, Gyeongju 38066, Korea; seodi74@dongguk.ac.kr; 5School of Sport Science, Pusan National University, Pusan 46241, Korea; jjparkpnu@pusan.ac.kr; 6School of Sport Science, University of Ulsan, Ulsan 44610, Korea; hanjoon@ulsan.ac.kr; 7Department of Sport Medicine, College of Natural Science, Soonchunhyang University, Asan 31538, Korea; violethjk@naver.com; 8Department of Preventive Medicine and Genomic Cohort Institute, Yonsei Wonju College of Medicine, Yonsei University, Wonju 26426, Korea; ysahn1203@gmail.com

**Keywords:** firefighter, exercise training, number of fire dispatches, situational factors

## Abstract

According to previous research, participation in exercise training (ET) by South Korean firefighters varies with shift type, and the effect of shift type is greater in large cities than in small towns. However, shift types differ among regions, depending on the number of dispatches. Therefore, the present study examined the impact of the number of fire dispatches and other situational factors on ET. A series of multilevel logistic regression analysis was applied to analyze the data collected from South Korean firefighters (N = 5219) in 2017. According to the firefighters, participation in ET is higher among those who have someone to instruct their ET (Coefficient (Coef) = 0.057, SE = 0.017, *p* < 0.001) and who can do ET while on duty (Coef = 0.048, SE = 0.014, *p* < 0.001). The number of fire dispatches had a significant effect on participation in ET (Coef = −0.000, SE = 0.000, *p* < 0.01), meaning that the firefighters’ participation in ET varies with the number of fire dispatches in each region. Our main findings indicate that the number of fire dispatches is a key factor affecting ET participation among firefighters, and the other situational factors also play a role. Therefore, ET programs that firefighters can participate in between calls should be established.

## 1. Introduction

Firefighters are required to have high levels of physical fitness compared with other occupational groups because they perform arduous tasks in strenuous environments [1,2]. Likewise, physical fitness is clearly related to their performance in fire suppression because they need to rescue victims while carrying approximately 22.68 kg of protective equipment [3,4]. However, according to previous studies, South Korean firefighters do not meet these physical fitness requirements [5,6]. Moreover, the level of participation in the sports activities of female firefighters is lower than in male counterparts in terms of frequency and intensity [7]. This may be because exercise training (ET) in South Korea is not mandatory, whereas countries such as the UK and the U.S. require their firefighters to participate in ET [8,9]. In particular, firefighters in the UK are required to maintain their physical fitness levels to thereby perform public services and to protect their colleagues and themselves. Otherwise, employers or employees could be liable to be prosecuted for violation of their duty under health and safety laws [9]. However, there is no such law in South Korea, even when firefighters do not maintain acceptable physical fitness levels. This could mean that, in contrast to the UK system, South Korean firefighters’ physical fitness levels are managed at the personal level, but not the institutional or structural level [10]. Furthermore, previous research into firefighters’ physical fitness has mostly focused on the relationship between physical fitness and job performance or the physiological effect on workers from worksite activity [11,12,13]. Since voluntary ET participation of firefighters has not been investigated, finding important factors that may affect ET participation among firefighters can help develop a better ET promotion program and can protect both society and workers’ health.

Compared with other public servants, firefighters’ duties have unique characteristics in terms of the environmental and hazard-prone nature of their work, as well as the shift patterns within that work. In particular, due to their shift patterns, many firefighters are at risk of coronary heart disease and sudden cardiac death [2,3]. In South Korea, a three-shift system (three groups, two circuits) was first proposed upon the launch of the National Emergency Management Agency on 1 June 2004, as an improvement to the two-shift system (two groups, two circuits). However, the change to the shift system has led to a serious shortage of manpower. Hence, the official three-shift system used in 2009 includes 6 days (day–day–night–night–off duty–off duty), 9 days (day–day–day–night–off duty–night–off duty–night–off duty), 21 days (day–day–day–day–day–off duty–off duty–night–off duty–night–off duty–night–off duty–duty–off duty–night–off duty–night–off duty–duty–off duty), and 3 days (duty–off duty–off duty) [14].

Operation of the 3-day system depends on the number of dispatches across 17 regions in South Korea. It is put into operation only if there is less than one fire suppression, three rescues, and four first aid callouts within a day [15]. As of 2017, 97% of firefighters in South Korea were working under the three-shift system, whereas those in other nations such as Singapore, Germany, France, and the U.S. already operate under the 3-day system [14,16]. As a number of developed countries operate under the 3-day system, which requires 48 h of rest and 24 h of work, 69% of South Korean firefighters prefer the 3-day system [17]. Nevertheless, according to a previous study conducted by Park et al. [18], firefighters participate in less ET under the 3-day system than under the 21-day system. Furthermore, they reported a significant interaction between the effects of shift type and city scale in terms of participation in ET. The effect of city scale on participation in voluntary ET was greater under the 3-day system compared with the 21-day system. Specifically, the significant dependence of ET participation on shift type is observed in large cities, but not in small towns [18].

A recent study assumed that it was easier for firefighters in small towns to participate in ET because they are required to attend fewer dispatches [18]; however, the effect of the number of dispatches on firefighters’ ET participation was only speculated. Moreover, data used in that recent study did not include any information regarding the number of dispatches in 2017. Hence, the assumption that the number of dispatches affects firefighters’ ET participation remains to be validated. Therefore, in this study, we investigated whether firefighters’ participation in ET varies with the number of dispatches across 17 regions in South Korea. Furthermore, in the above previous study, only 3- and 21-day shift patterns were analyzed [18], but the 9-day system is the second-most widely used in South Korea after the 21-day system. Thus, the 9-day system should also be examined.

In the U.S., the National Fire Protection Association (NFPA) presented an exercise program for firefighters designed to improve their health via ET while on duty [19]. The program was organized by the “NFPA 1583 (standard for health-related fitness programs for fire department members)”, which was established to improve U.S. firefighters’ health and job performance [8]. Although Article 23, Section 2 of the Framework Act on Health, Safety, and Welfare of Fire Officers in South Korea states that “The head of the fire department must take measures, such as securing physical fitness facilities and spaces, to allow employees to take regular physical care, and can designate and facilitate time for exercise training during the day”, ET during office hours is quite difficult. Furthermore, as mentioned above, ET is not obligatory for firefighters [1,20]. Previous studies indicate that narrow spaces and old equipment present impediments to firefighters’ ET in the workplace [21,22,23], but most ET facilities in South Korea are old and have limited space. This is probably why many firefighters have difficulties with ET [1]. A “Peer Fitness Program” was established to help U.S. firefighters manage their health under guidance from coworkers [24,25]. However, there are no such programs in South Korea and an enforcement of ET among firefighters mainly depends on the heads of firefighting agencies [26,27], implying that support for ET is relatively poor compared to the U.S.

To summarize, the majority of previous studies emphasized the importance of situational factors, such as having ET instructors among colleagues and having an availability of ET while on duty. However, we lack sufficient research on the impact of such factors on South Korean firefighters’ participation in ET. Hence, we aimed to investigate whether other situational factors significantly influence South Korean firefighters’ voluntary participation in ET and the following hypotheses are proposed:
**Hypothesis 1** **(H1).**Firefighters who have ET instructors among their colleagues will participate in ET more than those who do not have ET instructors.
**Hypothesis 2** **(H2).**Firefighters who can do ET while on duty will participate in ET more than those who cannot do ET while on duty.

Despite the significant effects of shift type and city scale on ET participation reported in the abovementioned recent study, the authors ignored the influence of the number of dispatches on the shift types used in the 17 regions of South Korea. Therefore, the second purpose of this study was to assess the impact of the number of fire dispatches in each region on participation in voluntary ET using a multilevel logistic regression analysis and the third hypothesis is presented as follows:
**Hypothesis 3** **(H3).**A lower number of fire dispatches in a region is associated with a higher participation levels in ET.

Furthermore, we also included health-related variables such as drinking and smoking. The majority of studies examined the association between drinking and smoking and health outcomes of firefighters [28,29,30], but most of them did not investigate the relationship between such behaviors and ET participation among firefighters. By investigating the clustering of various health behaviors (“syndemic” nature of health behaviors) [31], researchers and practitioners can gain an understanding of whether other health behaviors can be regarded as indicators of ET.

## 2. Materials and Methods

### 2.1. Data

For the main study, we used data from an online anonymous survey carried out in 2017 by the National Fire Agency (used to be the Ministry of Public Safety and Security) of South Korea. The survey was conducted with the aim of improving the health and safety of firefighters based on their work-related characteristics. The number of target population was approximately 40,000 firefighters nationwide and the response rate was 24.53% (N = 9812). A sample size of 4386 was estimated by a priori power analysis using G *power 3.1 statistical software written by Franz Faul, University Kiel, Germany. The α-level and power were set at 0.05 and 0.95 respectively and it was two-tailed [32]. In this study, 5219 (53%) of South Korean firefighters were included in the data and they were all outside workers. Participants were notified in writing of the aim of the study, research plan, data confidentiality, and privacy. Those who were not willing to response to the survey did not participate in the study. The type of survey was a closed-ended questionnaire and mainly included questions on general and professional characteristics such as job type, shift–work circuit, shift work, and ET participation, preferences, and environment. On average, it took 45 min to complete the entire questionnaire. This study was approved by Dongguk University Ilsan Hospital Institutional Review Board (IRB No. 2017-08-014-001). We requested consent from the head of the National Fire Agency, who is a representative of the firefighters who participated in this study. We also obtained the number of fire dispatches conducted in the 17 regions of South Korea in 2017 following an information disclosure request.

### 2.2. Measures

In our study, participants were asked to report their gender, age, and education level for demographic variables. For education level, participants were asked to indicate whether they graduated from high school, college, university, or graduate school. Participants were also asked to indicate their health-related behaviors. They were asked whether they drink alcoholic beverages or not and if they did, they were asked how many times they drank in a week and how many glasses they typically drank in one sitting. Likewise, they were asked whether they currently smoked, had quit smoking, or if they had never smoked before.

For the shift types, participants were asked to choose which day systems they are currently on including day-duty, 2 days, 3 days, 5 days, 6 days, 9 days, and 21 days.

ET was defined as “physical activity that any bodily movement produced by skeletal muscles that require energy expenditure” [33]. To assess ET, the following questions were asked: “Do you participate in ET to improve fitness?” (yes/no), “How many times a week do you participate in ET?” (e.g., once a week), “For how many minutes do you perform ET at once?” (e.g., 30 min). The answers to the last two questions were multiplied together to represent the total minutes of ET participation in a week. Firefighters were defined as an active participator if they participated in ET for 150 min or more in a week [33].

Situational factors were measured by asking the following questions: “Is there anyone who can instruct you in ET among your colleagues?” (yes/no) and “Is ET available while on duty?” (yes/no).

The number of fire dispatches in the 17 regions of South Korea in 2017 was divided by 365 days to represent the mean of fire dispatches per day in each region.

The ET and independent variables were categorical variables and age and number of fire dispatches were continuous variables.

### 2.3. Statistical Analysis

As firefighters’ shift type depends mostly on the number of dispatches in each region, we conducted a multilevel logistic regression analysis using SAS version 9.4 (SAS Institute Inc., SAS Campus Drive, Cary, NC, USA) and Hierarchical Linear Model Version 6.08 (Scientific Software International, Lincolnwood, IL, USA). To investigate the effects of other situational factors and the number of fire dispatches on ET, data from 5219 South Korean firefighters were included in the analysis. We targeted firefighters working under the 3-, 9-, and 21-day systems. These were all outside workers (i.e., firefighting, first aid activity, and rescue), because outside workers perform more physically demanding duties than inside workers do. Similarly, only the number of fire dispatches was obtained, because all outside workers are spurred into action when fire calls come through, whereas first aid and rescue calls do not require all available staff. In the multilevel logistic regression analysis, shift type, other situational factors, and demographic variables were included in level one, and the number of fire dispatches in each of the 17 regions of South Korea were included in level two. Demographic and health-related variables, such as sex, age, education level, drinking, smoking, and shift type, were controlled in model 1. In model 2, other situational factors, including an ET instructor among colleagues and availability of ET while on duty, were controlled. Finally, the number of fire dispatches was controlled in model 3. These models represent the associations between the effect of the number of fire dispatches and other situational factors on ET participation in firefighters in the 17 regions of South Korea, at both the individual and the city levels. The full maximum likelihood estimation method was used to account for missing values.

## 3. Results

### 3.1. Descriptive Statistics

In Table 1, we summarize the participants’ demographics. The 5219 firefighters comprised 4775 (91.49%) males and 444 (8.51%) females. The mean age of the participants was 38.11 (±8.62) years. Among the firefighters, 420 (8.05%), 761 (14.58%), and 4038 (77.37%) worked under the 3-, 9-, and 21-days systems, respectively. Approximately 80% of firefighters participated in ET for 30–90 min, 9% for 90–120, and 11% for 120–150 min per week respectively, and almost half—2548 (49.27%)—of firefighters participated in ET for > 150 min per week. A total of 2353 (45.09%) firefighters responded that they had colleagues to instruct their ET, and 2910 (55.76) confirmed that ET was available while on duty. The mean number of fire dispatches per city was 43.94 (±76.02) and the number of fire dispatches per day in each region is listed in Table 1.

### 3.2. Participation in Exercise Training

The results of the multilevel logistic regression analysis are shown in Table 2. In model 1, sex (Coef = 0.190, SE = 0.022, *p* < 0.001), education level (college: Coef = −0.080, SE = 0.016, *p* < 0.001; university: Coef = −0.091, SE = 0.012, *p* < 0.001), and smoking status (Coef = −0.082, SE = 0.021, *p* < 0.001) significantly affected participation in ET, while shift type (9 days: Coef = 0.061, SE = 0.036, *p* < 0.1; 21 days: Coef = 0.047, SE = 0.027, *p* < 0.1) did not significantly affect participation in ET. In model 2, the ET instructors among colleagues (Coef = 0.057, SE = 0.017, *p* < 0.001) and availability of ET while on duty (Coef = 0.048, SE = 0.014, *p* < 0.001) significantly affected participation in ET. The 21-day system significantly affected participation in ET (Coef = 0.050, SE = 0.025, *p* < 0.05). In the final model, the number of fire dispatches (Coef = −0.000, SE = 0.000, *p* < 0.01) had a significant effect on participation in ET, suggesting significant differences in ET participation among regions with different numbers of fire dispatches.

## 4. Discussion

To the best of our knowledge, this is one of the first studies on the relationship between ET participation and the number of fire dispatches in 17 regions of South Korea using a multilevel logistic regression analysis. The main objectives of our study were to examine the effects of the number of fire dispatches and other situational factors on voluntary ET among South Korean firefighters. As we hypothesized, a significant influence of situational factors was found and, more importantly, ET participation was associated with the number of fire dispatches in each region. According to our results, male firefighters participated in more ET compared with their female counterparts. This can be explained by the gender inequality of fitness criteria in South Korea [34]. The criteria of firefighters fitness tests in South Korea is much more generous to female firefighters [34] and, therefore, female firefighters may participate in ET less actively than male firefighters [7]. We need stronger criteria to motivate female firefighters to improve their fitness levels [35] because, just like male firefighters, female firefighters are required to participate in physically strenuous tasks [5]. In contrast to the general expectations, college or university graduates engaged in less ET than did high school graduates, and smokers participated more than non-smokers. These results can be related to the job characteristics of firefighters. According to the National Fire Agency, the job stress of firefighters was higher than that of the general public [36]. The job stress can lead to mental reactions and physical changes and can cause excessive smoking [36]. Since there is evidence also showing that firefighters smoke more than the general public [37], the effect of smoking behavior on ET participation of firefighters can be seen from a different perspective. In the same vein, firefighting jobs do not limit education level [38], and training such as worksite activities are included in the duty regardless of individual academic background. Therefore, it can be seen as a different context from the impact of education level on participation in the physical activities of the general public. Moreover, firefighters under the 9- and 21-day systems participated in more ET compared with those under the 3-day system. This result is consistent with those of a previous study [18]. Firefighters under the 3-day system have 48 h of rest after 24 h of duty. Since physically strenuous occupations such as firefighters tend to be less motivated to engage in physical activity during their leisure time [39], they possibly spend this time with their family or friends instead of participating in ET because they have more leisure time than the other two day systems and consider ET to be a work-related activity [16].

Furthermore, firefighters who indicated that they have colleagues as ET instructors engaged in ET more frequently than those who did not. According to other research on the health and physical strength of firefighters, if one firefighter has an active lifestyle and a healthy body, their colleagues are more likely be healthy and active [24]. Additionally, of the factors that affect the outcomes of fitness programs for firefighters, “supervision” increases their efficacy. Specifically, this refers to the ET involvement of exercise trainers and coaches who are professional instructors, there to guide firefighters. Compared with “non-supervision”, this helps improve the efficacy of exercise and physical fitness [40]. Similarly, many previous studies have emphasized the importance of peer fitness programs, which positively influence others to adopt healthy behaviors such as ET. This also helps people to instruct their colleagues in ET [41,42,43]. Even though there is no such program for firefighters in South Korea, it is clear that firefighters would be more likely to participate in ET if they had colleagues who can instruct their ET. Thus, firefighting organizations should establish peer-led programs to help firefighters improve their physical fitness. In addition, firefighters who responded that ET while on duty is available participated in more ET than those who did not. According to previous research, firefighters tend to participate in ET more regularly when they have scheduled times on shift, rather than off duty [44]. It is important that firefighters participate in ET by themselves, but securing time and resources to support ET programs with appropriate facilities and equipment should also be preceded [45]. In contrast to South Korea, it is mandatory in the U.S for firefighters to participate in ET programs while on duty [23]. Therefore, fire department captains should designate time for ET to manage their employees’ physical fitness levels and adhere to the article of the Framework Act on Health, Safety and Welfare of Fire Officers regarding ET while on duty [20].

Finally, the most important finding of this study was that ET participation varies with the number of fire dispatches. Hence, the greater the number of fire dispatches in a region, the less engagement of firefighters in ET. Therefore, the finding that South Korean firefighters participate in more ET under the 21-day rather than the 3-day system should not be attributed to the effect of city scale. In fact, this result extends on previous work by showing that the number of fire dispatches is the key factor determining the shift type used in each region of South Korea, and this has a direct impact on participation in ET. The main contribution of this study is our analysis of the key factors influencing firefighters’ participation in ET, particularly situational factors that have not been investigated in previous studies. Furthermore, the results of this study are helpful for understanding the group characteristics of South Korean firefighters with respect to ET participation. There has been evidence that firefighters working in a fire department that provides well-organized health promotion programs are more physically active than those who do not [46]. Therefore, policymakers and practitioners should consider the number of fire dispatches when developing ET programs that enable firefighters to participate in ET during their spare time in-between calls. More specifically, firefighting organization in South Korea should establish a department for firefighters’ physical fitness management and overall health and safety just like NFPA 1583 in the U.S.

We found a relationship between the number of fire dispatches and ET participation and analyzed the effect of other situational factors. However, there are several limitations to this study. First, the use of self-reported responses to questionnaires may result in recall or response bias. Second, our results are not generalizable to other populations due to the unique characteristics of firefighting. Firefighters’ participation in ET is clearly related to job performance. Hence, their intentions regarding ET may be different from those of the general population. Third, as this is a cross-sectional study, we could not demonstrate causal relationships between the variables. Thus, the reader must exercise caution when interpreting these results. Fourth, we could not test validity and reliability of ET. The data we used in this study concerned the number of days in which ET was participated in during the past week and how many minutes of ET were performed at once and, therefore, we thought it better to categorize ET based on the physical activity guidelines suggested by the World Health Organization. Fifth, we did not ask the type of ET participated in. Future research should ask more specific questions about ET participation among firefighters. Finally, we only investigated two situational factors affecting firefighters’ participation in ET and they were assessed using self-report questionnaires. Therefore, in the future, researchers should investigate the role of other relevant situational factors and should consider implementing an intervention to examine the relationship between ET and the number of fire dispatches and other situational factors. Despite these limitations, we hope that our findings will contribute to the literature by suggesting important information regarding South Korean firefighters’ participation in ET.

## 5. Conclusions

It is important to understand the factors that affect firefighters’ participation in ET because their physical fitness directly affects public safety and welfare. Our findings indicate that the availability of someone to instruct ET and of ET activities while on duty increases the frequency of firefighters’ participation in ET. Furthermore, we report that the number of fire dispatches in each of the 17 regions of South Korea is a key factor affecting participation in ET among South Korean firefighters. This finding is meaningful because we have extended the results of a previous study by determining exactly which factors affect ET participation. Hence, it is necessary to develop ET programs with designated instructors that firefighters can perform while on duty and in-between calls.

## Figures and Tables

**Table 1 ijerph-17-05913-t001:** Characteristics of firefighters.

Characteristics	Category	N	%
Individual level (*N* = 5219)			
Sex	Male	4775	91.49
	Female	444	8.51
Age (Mean ± SD)		38.11 ± 8.62	
Education	High school	1000	19.61
	College	1738	34.08
	University	2291	44.92
	Graduate school	71	1.39
Drinking	Yes	3828	73.38
	No	1389	26.62
Smoking	Yes	1442	27.64
	No	3775	72.36
Shift-type	3 days	420	8.05
	9 days	761	14.58
	21 days	4038	77.37
Other situational factors			
Exercise Training (ET) instructors among colleagues	Yes	2353	45.09
	No	2866	54.91
Availability of ET while on duty	Yes	2910	55.76
	No	2309	44.24
Exercise training (min/week)	≥150	2548	49.27
	<150	2623	50.73
City level (*N* = 17)			
Number of fire dispatches per day (Mean ± SD)		43.94 ± 76.02	
Sejong		0.865	
Jeju		2.057	
Gwangju		2.528	
Ulsan		4.257	
Daejeon		5.460	
Pusan		7.147	
Jeonbuk		9.167	
Gangwon		10.290	
Daegu		11.731	
Chungbuk		13.920	
Seoul		24.008	
Chungnam		26.060	
Incheon		27.008	
Jeonnam		28.383	
Gyeongnam		31.271	
Gyeongbuk		32.569	
Gyeonggi		274.852	

**Table 2 ijerph-17-05913-t002:** Multilevel logistic regression analysis of ET participation among South Korean firefighters (*N* = 5219).

Characteristics	Model 1		Model 2		Model 3	
	Coef	SE	Coef	SE	Coef	SE
Individual level (*N* = 5219)						
*Intercept*	0.380 ***	0.038	0.306 ***	0.035	0.313 ***	0.036
*Sex*						
Male (ref)						
Female	0.190 ***	0.022	0.181 ***	0.021	0.180 ***	0.020
*Age*	−0.001	0.001	−0.000	−0.001	−0.000	0.001
*Education*						
High school						
(ref)						
College	−0.080 ***	0.016	−0.074 ***	0.015	−0.074 ***	0.015
University	−0.091 ***	0.012	−0.086 ***	0.019	−0.088 ***	0.012
Graduate school	−0.071	0.061	−0.068	0.061	−0.068	0.060
*Drinking*						
Yes (ref)						
No	0.025	0.018	0.021	0.018	0.022	0.018
*Smoking*						
Yes (ref)						
No	−0.082 ***	0.021	−0.084 ***	0.022	−0.084 ***	0.021
*Shift-type*						
3 days (ref)						
9 days	0.061	0.036	0.063	0.035	0.062	0.035
21 days	0.047	0.027	0.050 *	0.025	0.053 *	0.026
*Other situational factors*						
ET instructors among colleagues						
Yes (ref)						
No			0.057 ***	0.017	0.058 ***	0.017
Availability of ET while on duty						
Yes (ref)						
No			0.048 ***	0.014	0.047 ***	0.014
City level (*N* = 17)						
*Number of fire dispatches*					−0.000 **	0.000

Note: Coef = logistic coefficient; ref = reference category. * *p* < 0.05, ** *p* < 0.01, *** *p* < 0.001.

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
