# Peer review of "The Effects of Number of Fire Dispatches and Other Situational Factors on Voluntary Exercise Training Among Korean Firefighters: A Multilevel Logistic Regression Analysis"

_ijerph, 2020, doi:10.3390/ijerph17165913_

Round 1

Reviewer 1 Report

The article performs an examination of the impacts of the numbers of fire dispatches on other sociodemographic factors on exercise training in firefighters.

The study shows that the authors have a wide domain on the subject and have some work related to the topic. I only have minor comments for the authors to address.

The review of the literature and its foundation seems to me to be adequate. The authors cite important and updated works as well as on other research carried out in the setting of fire dispatches and the importance of exercise training. Please provide a hypothesis for your objectives.

Regarding the method, the authors use adequate and robust techniques in statistical analysis. I suggest the authors reference the creators of G*Power.

The results are presented clearly and are easy to follow. Please provide confidence intervals for each coefficient.

The discussion is also very comprehensive. Start this section with your main objective.

Finally, the authors should look for evidence to support their claims. There is evidence of

Significant coefficients in the association between exercise training and several ID. However, some of them do not provide significant variance. Please try to justify why these variables should be interpreted when the variance is close or even equal zero.

Reviewer 2 Report

Dear Authors,

The topic of the article is original.

I believe the topic of your study is original.

Paper is generally well written and balanced.

References to the consistent bibliography have been made.

I have the following detailed comments:

ABSTRACT

OK

TITLE                                                                                                

OK

INTRODUCTION
The introduction provides an adequate background.

The objectives of the present study are clearly explained.

You should better specify in FITT terms the mandatory exercise training (ET) in US and UK.

You can insert a sentence on the difference between males and females, which is, and then you reported in the discussion.

MATERIALS AND METHODS

Data

Please, specify the type of questionnaire (open or closed questions) and the method of administration.

Measures

Please, specify the type of physical activity.
Line 145: It is better to add the recommendation by World Health Organization

Statistical Analysis

Line 163-169: Please, Explain better why you chose 3 models. In discussion, it should be explained that there is no difference between the individual models.
If there are no particular differences, it is preferable to use model 3.

RESULTS

Ok

Although significant the regression coefficient is 0 please explain in practice what it means

DISCUSSION

Line 201: Please, add the reference to support

Line 209: Please, add the sentence to support

I approve of the submission after major revision.

Reviewer 3 Report

Dear,

You have written an interesting paper; however, some parts need to be addressed.

The abstract needs to be rewritten and the main results with p values added.

The introduction is clearly written, but in the end, you mention as you call them ‘’health-related variables such as drinking and smoking’’ but they are not mentioned in the introduction by not even 1 research and how previous research of drinking and smoking impacts shift work or exercise training. Add that to the introduction.

Line 125 – From how many? What is the percentage of the Firefighter population? ADD

Line 126-127 – Did you get written consent from the participants? It is not clearly mentioned? ADD

How many surveys were not correctly filled (return rate)? ADD

Line 137 – So they were asked- Do you drink? So If I drink 1 beer per week, I will be categorised as a drinker? How many units per day or per week? Be more specific (what was normative). Additional explanation about the questioner is needed – How many questions in total and per each segment need to be reported. The same for smoking. Also, did you ask what kind of exercise training they performed (weight training, running, basketball, etc.) If not this should be included in the limitations.

How were the questions formed – Likert scale from 1-5 or 1 to 7? Report and be more specific.

Only in the statistical analysis, you divide the firefighters I the outside workers and I suppose inside workers. This should be presented in the description of the sample.

What level of significance did take into account? Report

Is the number of dispatches per city reported per day? Add, so it is clear.

Be more clear about the practical implications of your study.

Kind regards

Reviewer 4 Report

Summary:

The current study uses a multi-regression analysis to evaluate the factors that affect participation in exercise training in a large cohort of firefighters in South Korea.  Over 5000 firefighters completed a survey that gathered data on demographic variables, work schedule, and participation in exercise.  In regards to exercise, the firefighters were asked to detail the frequency, duration, and intensity of exercise as well as the availability of exercise participation and instructors while on duty.  Using current physical activity guidelines, the authors determined that approximately 50% of the firefighters performed the recommended activity.  Having instructors and scheduled time to exercise were important variables in participation in exercise training.  Conversely, undergraduate education, non-smoking status, and number of calls (dispatches) lowered exercise training participation.

Comments, Concerns, and Suggestions:

  1. What is the difference between college and university in South Korea? These both post-secondary institutions that offer undergraduate degrees (i.e. Bachelor’s), correct?  If so, then having separate categories is not necessary as the educational level achieved is similar.

  1. In the Methods (line 137), the authors indicated that the participants were asked “whether they drink or not”. The reviewer assumes the question is specifically asking about alcoholic beverages.  If so, this should be clearly indicated. 

  1. Some clarifications are needed regarding the survey questions relating to exercise training. Some of these answers should be added to the methods and/or included as limitations of the study in the discussion section. 
    1. In lines 142-143, one of the questions is, “For how many minutes do you perform ET at once”? Presumably, this is referring to continuous exercise.  If an individual performs intermittent activity for 10 minutes for a total of 40 minutes, would the data be reported as 10 minutes or 40 minutes?
    2. In lines 143-144, one of the questions is, “At what intensity do you perform ET?” Were the participants given examples in order to help them determine the appropriate intensity?  Obviously, capturing these data could be difficult without proper context.
    3. In the survey, does “exercise training” encompass both aerobic and resistance training? Is it possible that some of the firefighters perform weight training regularly with limited aerobic training and were characterized as low participation?

  1. The authors selected the ET categories of >150 minutes and <150 minutes based on physical activity guidelines. Perhaps, there would be interest in the data further categorized, assuming the data is available.  For example, what percentage reported exercise of 120-150min, 90-120min, etc.  In addition, the frequency (days per week) might be of interest.  

  1. In the submission, Table 1 is currently split over two pages. This needs to remain on one complete page.

  1. As a point of clarification… the reviewer is assuming that the firefighters in the study are paid and not volunteer. This is important.  In the US, approximately 2/3 of firefighters are volunteer are not necessarily subjected to the same physical fitness requirements as paid firefighters.

Round 2

Reviewer 2 Report

Dear Authors,

I appreciate changes you have made in the manuscript.

I think that it could be interesting for our readers in its current form. After revision, the paper has consistently improved in quality.

I approve of the submission after minor revision.

ABSTRACT

OK

TITLE

OK

INTRODUCTION
OK

MATERIALS AND METHODS

Data

OK

Measures

Line 158: health-related

Line 174: replace region with regions

Line 178: replace region with regions

Statistical Analysis

ok

RESULTS

Line 202-203 please reshape the sentence

DISCUSSION

Line 241 insert comma after since

Author Response

Dear Authors,

I appreciate changes you have made in the manuscript.

I think that it could be interesting for our readers in its current form. After revision, the paper has consistently improved in quality.

I approve of the submission after minor revision.

ABSTRACT

OK

TITLE

OK

INTRODUCTION
OK

MATERIALS AND METHODS

Data

OK

Measures

Line 158: health-related

Thank you for the comment. We have added hyphen between the words. It is yellow-highlighted.

Line 174: replace region with regions

Thank you for the comment. We have corrected the word. It is yellow-highlighted.

Line 178: replace region with regions

Thank you for the comment. We have corrected the word. It is yellow-highlighted.

Statistical Analysis

ok

RESULTS

Line 202-203 please reshape the sentence

Thank you for the comment. We have revised the sentence. It is yellow-highlighted in line 202-203.

DISCUSSION

Line 241 insert comma after since

Thank you for the comment. We have added comma after since. It is yellow-highlighted.

Reviewer 3 Report

Dear Authors,

After reviewing your corrections, I find the paper suitable for publication.

Kind regards

Author Response

Thank you very much.

This manuscript is a resubmission of an earlier submission. The following is a list of the peer review reports and author responses from that submission.

Round 1

Reviewer 1 Report

Thank you for conducting research on these important safety-sensitive shiftworkers. Understanding how working time impacts one's ability to accommodate other parts of one's life has a line of research spanning over 30 years. Some of the more recent research was reported on last year in the journal Industrial Health (https://www.jstage.jst.go.jp/browse/indhealth/57/2/_contents/-char/en). 

Regarding the introduction, I found it wordy, using a nomenclature I was unfamiliar with when describing the shift regime. For example, replacing the work "circuit" with "day" would help the reader understand the different rotas your are referring to. Some of the word usage was a bit hard to understand as well. For example line 49, "environment level", or line 65 "advanced" - meaning High Income Counties? I also felt that some of the assertions were a bit of a stretch, for example line 95 "...there are no such programs in South Korea implying that support for ET is relatively poor compared with in the U.S." Maybe, but is there any research to support that statement? Maybe the needs are met some other way? Is the statement even necessary given the paper does not probe much on it and the introduction was longer than necessary; the summary at the end (lines 98-106) provides much of the needed information. 

Regarding methods, I did not see whether the survey instrument or its items (especially the ones speaking specifically to ET) was tested or validated for reliability or validity. There was not actual measures of physical health or strength it seems, leaving it all to self-reporting, which has an increased chance of error that was only briefly touched on at the end of the discussion. 

Given the large sample size, I feel it is important for you to conduct a power analysis and modify the significance level accordingly. In addition, I am not a believer in something being marginally significant, the significance criteria is a cut-off. A result with a p<.1 is not significant. You also included a number of variables in your models - for example education level, smoking, etc (table 2), without providing a rationale for including them in the introduction, making the analysis look for exploratory than theory-based. 

For the discussion, I think the power analysis may change what you find as significant, for example shift type would likely fall out. I also felt that some of your statements jumped to conclusions without much substantiation, either in this section or in the introduction. For example, 187 "Male firefighters may participate in ET more actively that female firefighters. However, we need stronger criteria to motivate female firefighters to improve their fitness levels because..." - there is nothing in this study that looks at motivations of individuals. There is literature about gender role differences and how there are different expectations for women and men in different cultures, but none of that has been explored in this paper until this statement. A little later on - line 194, you provide a possible explanation of not exercises because the firefighters want to spend time with family and friends. Again, there is a lot of literature on this topic, uncited here, but you have little data to make this assertion from what you presented in the paper. 

In the end, you conclude that the more fire calls, the less ET (and opportunity for it), and if an ET program/trainer is present, respondents claimed they would be more engaged in ET. I feel that what is needed is to test this by providing an intervention and seeing if people respond as they claimed they would, while balancing the rest of their work/life, and if it can be sustained. It is hard to place the current results within the cited literature and make sense of it given it is all self-report.

Reviewer 2 Report

  1. The introduction section should be a little more theory-driven. Please present more rationale on how your study contributes to current literature on examining the impact of exercise training on the overall society and not only Korean firefighters. Overall, it seems that the Introduction section failed to present a convincing rationale as to why this study is needed, and what the benefits will be both to science and society in public health-related contexts. This is my main concern – I do not think the research question is innovative and relevant enough and what practical implications could emerge from this study.
  2. The authors state having recruited 40.000 firefighters. But in the result section, they only refer to n = 5219. It could be a typo but this question the lack of coherent research.
  3. Line 121-134: this is not an indication of a measure. This should not be reported in the method section.
  4. The authors in their title state using a multilevel approach. However, in the statistical analysis, they report using a hierarchical linear regression model. In the results, they state “multilevel logistic regression”. So, what did the authors? It seems that you did a step-wise logistic regression analysis using mostly binary independent variables (except for age). Using different names confuses the reader and cast doubts on the author's own understanding of their analyses. Please revise.
  5. There is no evidence that this study was approved by an ethical committee, nor did it follow the Helsinki declaration.
  6. Did the potential participants disclose informed consent?
  7. Where are the odds ratio estimates?
  8. Where are the confidence intervals?
  9. Where is the Nagelkerke’s R squared?
  10. Where is the ROC curve for age?
  11. Did the authors conduct apriori power analysis to work out your desired sample size? If no why not?
  12. How can a coefficient of “-0.000” be significant at p>0.001? I have never seen a regression coefficient in a hierarchical logistic regression model (in fact in any model at all) a value of -0.000 be significant. Please revise your data or argue about why this happens.
  13. The authors should avoid causal language when conducting tests with cross-sectional data. (e.g., line 228, line 232)
  14. Line 188: Bold statement. Needs reference
  15. The discussion section is extremely descriptive. You are making some very general sweeping statements with very little academic of theoretical considerations.
  16. I am not 100% sure how the measure you have used can provide practical implications for professionals and scholars – if you are going to make such arguments then much more evidence and theoretical discussions are needed.
  17. I would like to stress out that the authors state, “Nevertheless, according to a previous study conducted by Park et al [14], firefighters participate in less ET under the 3-circuit than 21-circuit system”. and cite a study by Park. et al. [14] in support of this. I want to point out that this appears to be the on authors, also using a circuit system in a sample of firefighters, though the citation given provides a non-essential page reference (see #34). I am concerned that the authors are essentially citing the present study as evidence of itself. I invite the editor to comment on this.
  18. Related to this, it appears that the authors systematically consider their own work to support their claims and arguments. Reference 5. I invite the editor to comment on this.
  19. It is worth to mention that the English writing in the current manuscript is not for publication-quality and requires major improvements. There are frequent grammatical and typographical errors present throughout the paper, which mar the author's own understanding of the research.